

# A High Duty Cycle Transmitter Unit for Steady-State Surface NMR Instruments

Nikhil B. Gaikwad [1], Lichao Liu [2], Matthew P. Griffiths [2], Denys Grombacher [2], and Jakob Juul Larsen [1]

[1]Department of Electrical and Computer Engineering, Aarhus University, 8200 Aarhus N, Denmark
[2]Hydrogeophysics Group, Department of Geoscience, Aarhus University, 8000 Aarhus C, Denmark

**Correspondence:** Nikhil B. Gaikwad (nbg@ece.au.dk)

**Abstract.** Groundwater measurements using surface nuclear magnetic resonance (NMR) have been notoriously challenged by a poor signal-to-noise ratio (SNR), but a new steady-state methodology based on long, high duty cycle, phase-locked pulse trains have demonstrated huge SNR increases. The hardware requirements for transmitters for steady-state surface NMR are significantly increased compared to transmitters for standard surface NMR use, due to the need for very high pulse-to-pulse

stability over long survey times and the increased thermal load caused by a much higher duty cycle. Further, the increased SNR leads to increased production rates, which necessitates lightweight equipment, that can be carried easily between many field sites during surveys. Here we demonstrate a novel steady-state surface NMR transmitter with a maximum 93 A peak current. The stability of the transmitter is evaluated on 10 minutes, 10% duty cycle, pulse trains containing pulses of either 5 ms, 10 ms, 20 ms, or 40 ms duration, and low or high current. We observe less than 150 ns pulse-to-pulse timing jitter and amplitude

variations below 0.4% between pulses for all pulse durations and currents. During tests, we observed no temperature effects on the timing and current stability. We have designed a customized heatsink, which reduces the transmitter weight by 30% and size by 16% without compromising safe thermal operating conditions. We evaluate the capacitor bank size and current stability and demonstrate that a 10 mF capacitor bank is an appropriate trade-off with insignificant current drooping in measurements. The extensive analysis and verification demonstrate that the transmitter generates highly stable pulse trains resulting in high

SNR signals.

## 1   Introduction

Groundwater is the most important source of water for billions of people in the world, but the resources are threatened by over-exploitation, pollution, and climate change (Postel, 2000; Famiglietti, 2014; Liu et al., 2017). There is a need for non-invasive, geophysical tools that can be used in exploration of new aquifers and in monitoring of existing aquifers.

Many electric and electromagnetic methods are used for this purpose (Binley et al., 2015; Wu et al., 2019; Beauchamp et al., 2018). One well-established method is transient electromagnetics (TEM), which is used to measure the electrical resistivity of the subsurface as a function of depth. TEM instruments can be mounted on moving platforms, both ground-based and airborne, and can therefore provide 2D and 3D maps of the subsurface resistivity (Siemon et al., 2009; Wu et al., 2019). Due to the correlation between the electrical resistivity and the subsurface lithology, e.g., clays have low resistivity and sands have high





resistivity, geological structures in the subsurface can be mapped and potential aquifer sites can be located. The electrical resistivity is affected by the presence of groundwater, but is not directly sensitive to water. Interpretation of data can therefore be ambiguous. The only geophysical method with a direct sensitivity to groundwater is surface nuclear magnetic resonance (NMR) (Behroozmand et al., 2015).

Surface NMR is based on the same principles as magnetic resonance imaging. In a static magnetic field, such as the earth

magnetic field, the nuclear spins of hydrogen nuclei are preferentially aligned along the magnetic field. After the nuclear spins are perturbed by an electromagnetic pulse at the resonant frequency, they relax back to equilibrium and emit an NMR signal, which can be detected with an induction coil. The amplitude of the NMR signal is proportional to the number of excited nuclear spins, i.e., the water content. The relaxation time of the NMR signal is controlled by the host material porosity and therefore carries information about sub-surface constituents and the ability of water to flow (Legchenko et al., 2002; Behroozmand et al.,

2015). A surface NMR measurement is conducted using a large coil, typically between 50 m × 50 m and 100 m × 100 m, laid out on the surface, to transmit the NMR pulse and receive the NMR signal. Data are collected using a sounding approach where the pulse current controls the effective depth of excitation. By repeating the measurements with different pulse currents, the water content and relaxation times are mapped as function of depth.

NMR signals from groundwater are very low amplitude, with amplitudes in the ten to hundred nV range. Consequently,

signals are easily drowned in electromagnetic noise from infrastructure or atmospheric sources (Kremer et al., 2022). Since the invention of surface NMR in the early 1980's a significant amount of research has been concerned with improving the signal-to-noise ratio (SNR) either from the hardware perspective or from the signal processing perspective. Focusing on the hardware side, multichannel instruments capable of remote reference noise cancellation has been very successful (Radic, 2006; Walsh, 2008). Other works have focused on reducing the instrument dead time to capture the early high-amplitude part of decaying

signals (Walsh et al., 2014; Li et al., 2015; Du et al., 2020; Lin et al., 2020), on hardware for specialized application in tunnel construction (Costabel, 2019; Yi et al., 2019), on receiver coils geometries with reduced noise sensitivity (Girard et al., 2020; Wang et al., 2022) or on increased depth of investigation (Legchenko and Pierrat, 2014). Adiabatic NMR excitation, where the frequency and amplitude are modulated during the excitation pulse, can provide a more uniform excitation of the subsurface and hereby enhance the NMR signal amplitude, but at the expense of depth resolution (Grunewald et al., 2016). Recently,

signal enhancement through pre-polarization of the nuclear spins by a strong DC field has been introduced, but the depth of investigation is limited to less than one meter (Lin et al., 2021b; Hiller et al., 2021). It is noteworthy that the development of surface NMR transmitters has mainly taken place at commercial companies and available publications are often sparse on technical details, e.g., (Radic, 2006; Walsh, 2008).

The standard surface NMR experiment is the free induction decay (FID) where the relaxation following a single transmitter

pulse is measured (Jiang et al., 2021). To ensure a high signal-to-noise ratio, the experiment is repeated many times over and the data are averaged. However, to ensure that the NMR experiment starts from equilibrium conditions, which is an implicit condition in standard modeling codes, it is necessary to include a wait time of typically about 5 s between pulses, which leads to very long acquisition times when averaging of tens or hundreds of pulses for each of typically ten to twenty different pulse currents are used.





We have recently introduced a novel steady-state approach to surface NMR, which is radically different from FID measurements (Grombacher et al., 2021, 2022). Instead of single pulse excitation, we use a train of closely spaced, $\sim 100$ ms, identical pulses to drive the nuclear spins into a new equilibrium, the steady-state, which is measured between pulses. This eliminates the wait time and allows us to measure much faster, i.e., approximately ten times per second in contrast to the standard once every five seconds. This fiftyfold increased data rate and new narrowband signal detection methods lead to orders of magni-

tude enhancement in SNR and in turn allows for much faster measurements and hence enables a much higher production rate, where many different groundwater targets can be measured in a single day. Steady-state methods are a well-developed concept in other fields of NMR and dates back to the 1950'ies, (Carr, 1958), but steady-state methods have until now not been applied to surface NMR as the necessary transmitter technology did not exist.

    The new steady-state pulsing scheme places much higher demands on the NMR transmitter electronics than standard FID

measurements. The steady-state concept enforces very strict requirements on the amplitude stability and the timing jitter between pulses and the continuous pulsing leads to a higher thermal load on the instrument. Further, the improved signal-to-noise ratio leads to a much higher production rate, i.e., more field sites per day, and the weight of the instrument becomes an important parameter.

    The aim of this paper is to discuss the specific requirements for a high duty cycle steady-state NMR transmitter, present our

designs, and document the electronic and thermal performance of the transmitter. In particular, we demonstrate the amplitude and timing stability of the pulse trains over extended measurement times, which is key to achieving the NMR steady-state. The paper is structured as follows. In the next section, we discuss the requirements of the transmitter from the NMR physics perspective. This is followed by a discussion of the design of the transmitter, simulations of the instrument, and experimental verification before concluding remarks are given.

## 80  2   Challenges in Steady-State Surface NMR

The concept of steady-state surface NMR is illustrated in Fig. 1. The transmitter delivers a continuous train of closely spaced NMR pulses. The spacing is so close, that the nuclear spins do not return to the original equilibrium state between pulses. Instead, after a few seconds of pulsing, the nuclear spins enter a new equilibrium, the steady state, where excitation balances relaxation.

By varying the pulse train parameters, e.g., pulse duration or delay between pulses, different steady-states are obtained, and the properties of the groundwater can be extracted from data using numerical modeling of the NMR Bloch equations (Griffiths et al., 2021). Contrary to common belief (Zhu et al., 2019), the envelope of an NMR transmitter pulse does not have to be boxcar shaped, neither for FID or steady-state. If the full waveform of the NMR pulse is measured, the pulse shape can be included in the modeling (Griffiths et al., 2021). The defining aspect of the steady-state pulse train is that each pulse is identical, i.e.,

current drooping within pulses is a non-issue, whereas current drooping between pulses will prevent the nuclear spins from reaching a steady-state. Conventional transmitters that do not maintain consistent inter-pulse amplitudes are therefore incapable



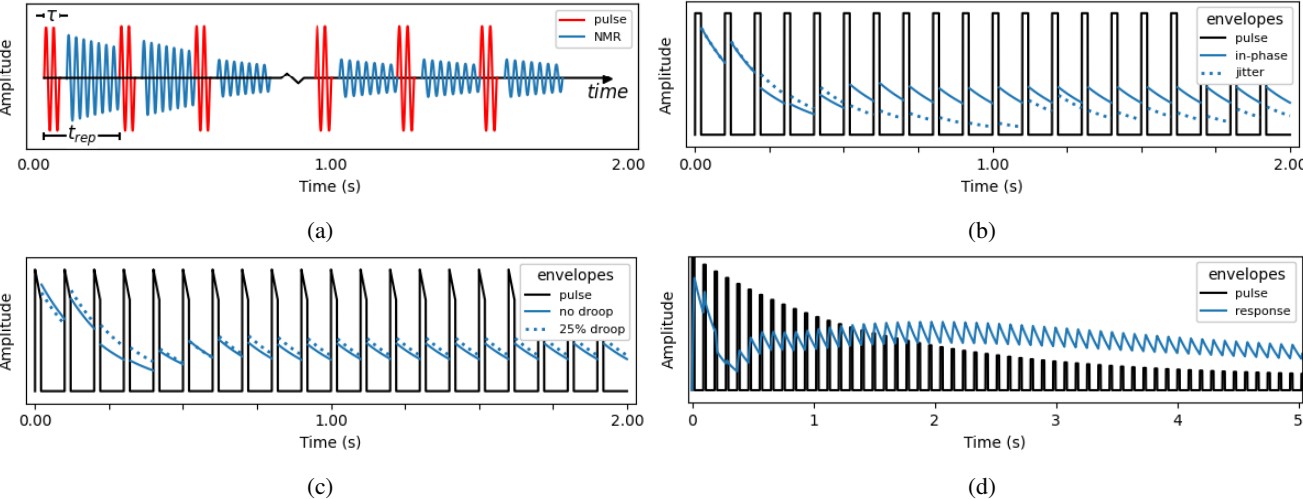

**Figure 1.** Steady-state surface NMR physics. a) Illustration of a steady-state sequence with pulse duration, $\tau$, and repetition time, $t_{rep}$. After an initial transient period of a few seconds, the magnetization reaches a periodic steady-state. b) Phase coherence, i.e., pulse timing, is critical for achieving and sustaining steady-state magnetization. The envelopes of two magnetization responses overlie the envelope of the pulse train (black). The full blue curve shows how a steady-state is reached with phase coherent pulses. The dotted blue curve show how a pulse train with jitter can not reach a steady-state. c) Energy loss of the transmitter can occur, particularly for long and rapid pulses, resulting in amplitude droop during the pulse. Here each pulse in the pulse train (black) exhibits a 25% reduction in amplitude by the end of the pulse. Despite this energy loss, the magnetization achieves a periodic steady-state (dotted blue curve). The full blue curve is the magnetization response to constant amplitude, in-phase pulses. d) Stable pulse amplitudes are also critical for sustaining a steady-state response. Here diminishing energy from pulse to pulse prevents a steady-state from occurring.

of retaining a steady-state. Likewise, the timing jitter between transmitter pulses should be less than a few percent of the NMR oscillation period, to ensure phase coherence and build-up of the steady-state, Fig. 1.

## 3 Transmitter design

### 3.1 System Overview


Our surface NMR instrument, denoted Apsu (Liu et al., 2019; Larsen et al., 2020), shown in Fig. 2, includes four main subsystems. The instrument is powered from a 1 kW generator that feeds into the 0-600 V controllable power supply (ApsuPS). The NMR pulses are generated by an H-bridge in the transmitter unit (ApsuTX) with the pulse waveform controlled by the voltage and switching timing of the H-bridge (Larsen et al., 2020). The current pulses are injected into a large coil, typically

50 m × 50 m, to produce the NMR pulses. The ApsuTXC is the main unit in the instrument. It contains the circuitry that sequences the H-bridge control signals. An internal 200 MHz clock is used to control the on/off timing of each individual half-cycle oscillation in the pulse train. The unit also contains the receiver part where the NMR signal and the current pulse



waveform is sampled at 31.25 kHz with two 24-bit A/D converters (Liu et al., 2019). The analog output from the high-bandwidth current probe (ApsuCP) is forwarded to the ApsuTXC where it is sampled. The ApsuCP also contains the circuitry

for operating the large coil in coincident mode, i.e., the same coil is used to transmit the NMR pulses and receive the NMR signals.

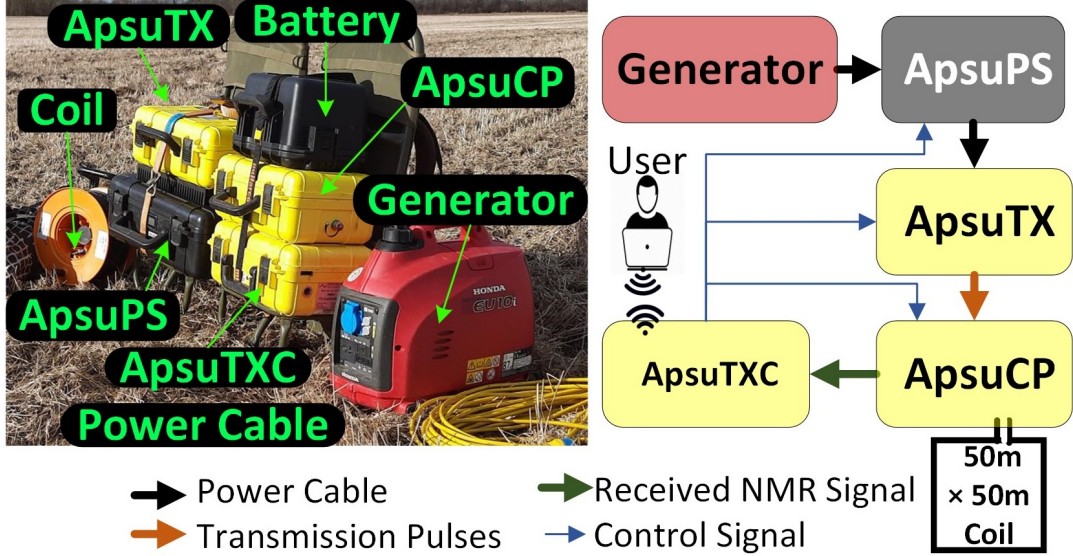

**Figure 2.** Apsu surface NMR instrument mounted on two backpacks and schematic overview of the Apsu workflow.

The first version of the ApsuTX was designed to run conventional FID sequences with relatively low power requirements (Larsen et al., 2020). With steady-state sequences, the power requirements are significantly increased. We use different pulse sequences to generate different steady states. Typically, a pulse sequence lasts 60 s, which ensures that the initial few seconds

of data where the steady-state builds up can be culled and enough data are left for retrieval of the NMR signal. In standard surveys, we often use more than 30 different pulse sequences. Consequently, the instrument is in continuous operation for more than 30 minutes with a pulse duty cycle around 10%. In addition, our surface NMR instruments are commonly used for remote surveys across the globe, often in challenging conditions. Therefore, the instrument needs to be compact, light in weight, handy, and efficient. These challenges motivated us to revisit our instrument for improved designs, especially the transmitter unit and

its heavy heatsink.

### 3.2 Thermal Management for ApsuTX

The heatsink is responsible for dissipating power lost during transmission. The H-bridge and damping resistor ($R_d$) are the main contributors and are physically mounted on the heatsink. In the first version of the ApsuTX, a commercially available standard heatsink (SH) was used. During this research, a thermal model of the transmitter was developed and used to design a

customized heatsink (CH) to ensure as low weight as possible while maintaining the necessary cooling properties. The thermal

**Figure 3.** Thermal model of the ApsuTX considering H-bridge and damping resistor ($R_d$) as primary thermal sources.



flow in the ApsuTX is simulated end-to-end to and the simulation is used to decide the appropriate size of the heatsink. Figure 3 shows the thermal model of the ApsuTX where the MOSFETs have been chosen to achieve fast H-bridge switching time (Lin et al., 2021a). The thermal power dissipation of MOSFETs mainly depends on drain-source ON resistance, peak current, and switching frequency and these device parameters are considered in the simulation(Chen et al., 2020). The model provides

access to the junction temperatures $T_j$ of the MOSFETs, which must be maintained at less than $150°C$ for stable operation. The junction temperature of the MOSFETs are given by

$$T_j(t) = \int_{0}^{T_p} P(t) \times \frac{\mathrm{d}}{\mathrm{d}t} Z_{th}(T_p - t) dt \tag{1}$$

where $P(t)$ is the electric power dissipated by a MOSFET and $T_p$ is the total time of the pulse (B.V., 2021). The thermal behavior of all four MOSFETs devices is modeled using the first order Foster thermal model (Schutze, 2008). The transient

thermal impedance ($Z_{th}$) of the Foster thermal model is expressed as

$$Z_{th}(t) = \sum_{i=1}^{n} r_i \times \left(1 - e^{-\frac{t}{\tau_i}}\right), \tag{2}$$

where $r_i$ is thermal resistance, $\tau_i$ is the time constant and $n$ is order of the model. The time constant is given by $\tau_i = r_i \times c_i$, where $c_i$ is the thermal capacity of the device (B.V., 2021), see Table 1. The thermal power flow through a MOSFET ($P_{th}$) is calculated using

$$P_{th}(t) = \frac{\triangle T}{r_i} + c_i \frac{\mathrm{d}\triangle T}{\mathrm{d}t}. \tag{3}$$

Here, $\triangle T = T_j - T_c$, with $T_c$ being the case temperature of the device. All MOSFETs and $R_d$ are mounted with thermal paste as an interfacing agent on the heatsink. The model considers the effect of the thermal paste, where its thickness, area, and thermal conductivity constant decide the thermal conduction rate. Also, the power dissipated by $R_d$ and transferred to the heatsink through thermal resistance is included in the model. The heatsink dissipates the thermal energy to the surroundings

through thermal convection, which depends on the material properties and the surface area. The model allows us to test the instrument under extreme conditions such as high ambient temperatures, high current, and high duty cycle for different pulse sequences.

## 3.3   Current Drooping

Consistency in pulse amplitudes is one of the critical concerns in steady-state surface NMR transmission. The power for the

high-current pulses is supplied from the ApsuPS that contains a 0-600 V DC supply unit and the capacitor bank. The ApsuPS unit is powered by a 1 kW generator, which provides 1 A output. A lightweight generator and ApsuPS have been chosen for field survey reasons, but it limits the current capacity of the power supply. This results in current drooping in the initial part of





the high current pulse train (See Figure 8b). Because of this inconsistent amplitude, the initial portion of transmission pulses cannot be used to reach the steady-state. The current drooping is usually seen because of internal impedance present in the

power source. This amplitude drooping cannot be removed entirely because of instrument limitations, but it can be optimized by selecting the correct capacitance value. The capacitance value, the drooping time, and the energy per pulse are interdependent. The drooping time and energy per pulse increases with a rising capacitance value. Ideally, the drooping time should be as short as possible and the energy per pulse as high as possible. However, both requirements can not be fulfilled simultaneously. The capacitance value must be selected as a trade-off between these interdependent parameters, which can also indirectly help to

reduce the size and weight. As noted above, the NMR steady-state is easily reached, regardless of any current drooping within transmitter pulses, as long as the current drooping is identical from pulse to pulse.

### 3.4 Pulse-to-Pulse Timing Stability

Like amplitude stability, pulse-to-pulse timing stability also plays a crucial role in reaching the NMR steady-state. The stability must be maintained throughout the entire pulse train, typically about one minute and containing several hundreds of NMR

pulses. The oscillation frequency of each NMR pulse, the Larmor frequency (Behroozmand et al., 2015), is matched with the local Earth magnetic field. Assuming a 2 kHz Larmor frequency, the timing jitter between transmitter pulse must be less than 1.4 $\mu$s to ensure less than $1°$ jitter in the pulse phase. The instrument clock is derived from a GPS disciplined oscillator, which controls the onset of each pulse in the train. The timing of each half-cycle within each pulse is controlled by an internal 200 MHz clock. It is necessary to control the pulse-to-pulse spacing so that there is an integer number of periods at the

Larmor frequency between pulses to ensure a coherent excitation of the NMR instrument. For optimum resolution of the NMR instrument parameters, i.e., water content and relaxation time as function of depth, the NMR instrument should be driven to many different steady states. This is achieved by varying the parameters of the pulse train such as the pulse-to-pulse delay and the duration of each pulse. It is also beneficial to use more advanced pulse trains, e.g., with the phase alternating between pulses or with an off-resonance frequency (Grombacher et al., 2022).

## 4 Simulation Method and Experimental Setup

### 4.1 Thermal Simulation for Two Different Heatsinks

The MATLAB-Simulink simulation starts from an older design, with a standard heatsink. All required thermal paste and heatsink dimensions have been measured and are included in the simulation. The second simulation is done for the custom heatsink with reduced dimensions. Both simulations are run for three different cooling techniques and four different sequences.

Typically heat transfer coefficients varies from 2.5 to 500 W/(m$^2$K) depending on the type of cooling. We have broadly divided this range into three types, where 5 W/(m$^2$K) corresponds to poor passive cooling, 10 W/(m$^2$K) to normal passive cooling, and 100 W/(m$^2$K) to active cooling. As ApsuTX is run with different pulse durations in the pulses sequences, e.g., 10 ms, 20 ms, 40 ms, and 100 ms, this is also included in the simulations.



**Table 1.** Model parameters for thermal simulations.

| Simulation Block | Parameter | Value |
|---|---|---|
| **H-Bridge** | Peak Current | 93 A |
| | Drain-source ON Resistance | 0.041 $\Omega$ |
| | Switch-on Loss | 0.0015 J |
| | Switch-off Loss | 0.00378 J |
| | Junction Mass | 0.1 J/K |
| | Initial Junction Temperature | 300 K |
| **Foster Thermal Model** | Thermal Resistance ($r_i$) | 0.104 K/W |
| | Thermal Time Constant ($\tau_i$) | 2 s |
| **Thermal Paste** | MOSFET's Area connected with the paste | 5760 mm$^2$ |
| | DR Area connected with the paste | 532 mm$^2$ |
| | Thickness | 0.2 mm |
| | Thermal Conductivity | 70 W/(mK) |
| **Standard Heatsink (SH)** | Mass | 3.5 kg |
| | Conduction Area | 60000 mm$^2$ |
| | Conduction Thickness | 8 mm |
| | Convection Area | 496000 mm$^2$ |
| **Customized Heatsink (CH)** | Mass | 1.3 kg |
| | Conduction Area | 60000 mm$^2$ |
| | Conduction Thickness | 8 mm |
| | Convection Area | 128000 mm$^2$ |
| **Heat Transfer Coefficient for SH and CH** | for Active Cooling | 100 W/(m$^2$K) |
| | for Normal Passive Cooling | 10 W/(m$^2$K) |
| | for Poor Passive Cooling | 5 W/(m$^2$K) |
| Specific Heat of | Aluminum | 903 J/(Kkg) |
| Thermal | Conductivity of Aluminum | 239 W/(mK) |
| Thermal Resistance | of Damping Resistor | 3 K/W |





## 4.2 Test setup of ApsuTX with Customized Heatsink

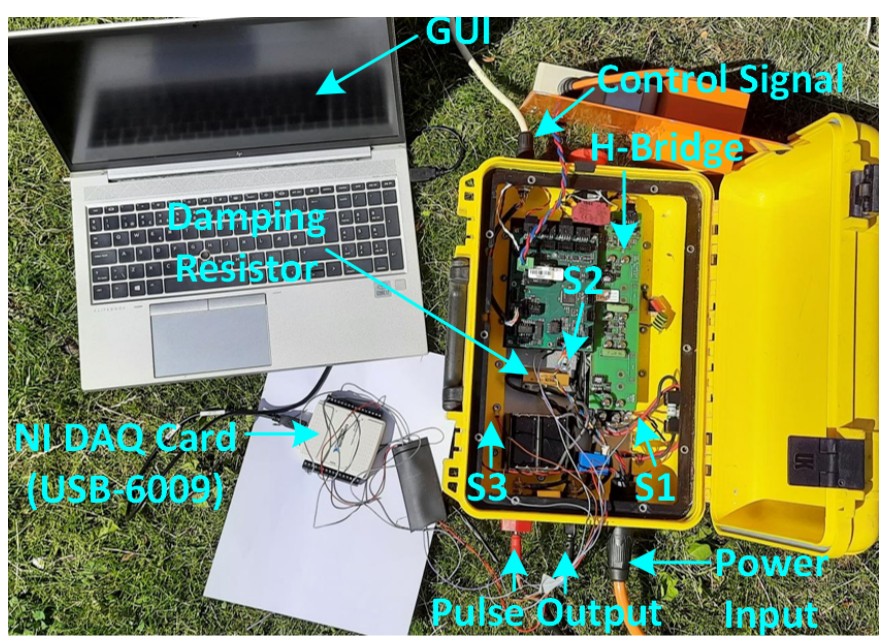

**Figure 4.** Experimental test setup to evaluate the thermal performance of the new ApsuTX in the field surveys. The labels S1, S2, and S3 indicate MOSFET, $R_d$, and heatsink.

After verification through simulations, a new customized heatsink has been fabricated and replaced the old heatsink. Figure 4 shows our field test setup. It includes three different temperature sensor nodes mounted on the MOSFET case (S1), the $R_d$ case (S2) and the customized heatsink (S3). Sensors S1 and S2 cannot measure the internal MOSFET junction temperature and $R_D$ temperature, but they are included to give a better understanding of the temperature variations in the transmitter. Sensor S3 directly measures the heatsink temperature. A National Instruments DAQ card collects these three analog voltage values

sensed by NTCALUG01A Series NTC Thermistors. The data are processed and time-stamped using a LabVIEW-based GUI. For the main field tests, we use steady-state sequences with 20 ms pulses and 72 A peak current, which are run continuously for 10 minutes. This is a much more intense sequence than used in the field where high current sequences are normally only 1 minute.

## 4.3 Current Drooping Simulation

The drooping simulation includes the DC supply with the limited current capability and a capacitor bank, which emulates the variable DC power supply. We have run simulations with nine different capacitor banks from 1 mF to 100 mF, to understand the effect of current drooping in the transmitted NMR pulses. The capacitor banks need different charging times to reach their maximum voltage output. The capacitor bank with 100 mF required the longest (75.5 s), and 1 mF required the shortest time





for complete charging (1.1 s). Also, the transmitted pulse generated by a higher capacitor bank is more stable than pulses from
the lower capacitor bank.

The simulation instrument transmits high current pulses with different intra-pulse and inter-pulse drooping. The time duration for amplitude stabilization and energy per pulse (after stabilization) is evaluated for all types of drooping. From the simulation results, we select the best suited capacitor value. The instrument is tested on the field, and the chosen capacitor bank value is validated by the observed drooping.

## 4.4 Pulse Stability Measurement Technique on Field Data

Collected data has been studied to verify the pulse stability. We measured pulse drifts for 5 ms, 10 ms, 20 ms, and 40 ms sequences. The pulse trains are sampled at 31.25 kHz by the ApsuTXC. The pulses are organized in the matrix with their respective time stamps. We fit a sinusoidal signal model to each pulse in the train, using nonlinear regression in MATLAB. The fitted signal measures the effective transmitted current through the coil at the Larmor frequency (Larsen et al., 2020), i.e., the component that must be phase coherent between pulses for steady-state operation. We validate the instrument performance by evaluation of the phase and amplitude of all pulses for two different currents and four different sequences.

## 5 Results

## 5.1 Thermal Performance of ApsuTX

A customized heatsink has been designed according to the simulations, field specifications, and compatibility with circuit placement. Figure 5 shows the temperature changes for both types of heatsinks, cooling conditions, and extreme test duration. Figure 5a presents the worst-case junction temperature ($T_J$) rise for both heatsinks and at different cooling conditions. It shows a 50 K temperature rise for 50 minutes of continuous pulsing at 93 A for the customized heatsink and poor passive cooling. Both heatsinks performed better for normal passive and active cooling. For the safe functioning of the H-bridge, the junction temperature must be maintained below 423.15 K, which is far from the observed temperature rise in the simulation. The simulations show that the junction temperature is maintained in a safe operating range even with poor cooling conditions in the ApsuTX with the customized heatsink. Active cooling shows better results, but active cooling is not feasible due to the increased size and weight.

Unsurprisingly the transmitter mounted with the larger, standard heatsink performs better than if it is mounted with the customized heatsink, but importantly, it can run within safe thermal management using either. As shown in Fig. 5b, $R_d$ is the most significant contributor to the thermal load. The cooling method and type of heatsink do not give rise to any significant differences in temperature variations. Finally, as shown in Fig. 5c, the heatsink temperature shows trends similar to the junction temperature. The plots show that the compact and lightweight CH can fulfill all thermal management requirements. The CH has been fabricated and mounted on the updated ApsuTX and we have measured the temperature at three key test points during field tests. Figure 6 shows the temperature data recorded during high current transmission at 72 A for 10 minutes.

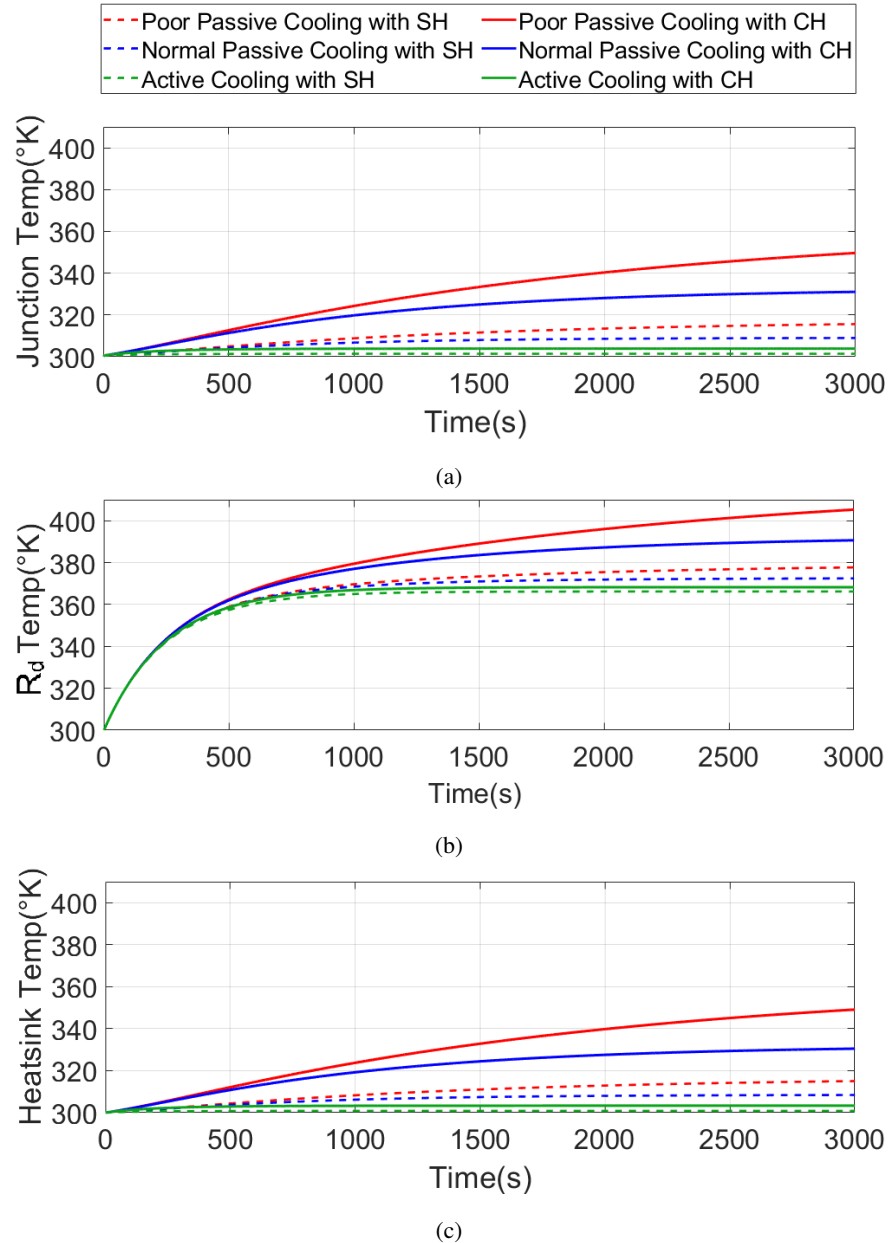

**Figure 5.** Effect of different cooling techniques on a) junction temperature ($T_J$), b) on $R_d$ temperature and, c) heatsink temperature for both SH and CH, running for 50 minutes.

Sensors S1 and S2 measure the temperature of the MOSFET case and $R_d$ case, which both continuously increase over time. We note that these test points do not give the internal temperatures of either component but provides a basic understanding of the temperature changes. The S3 sensor records the heatsink (CH) temperature. The temperature essentially follows the trend



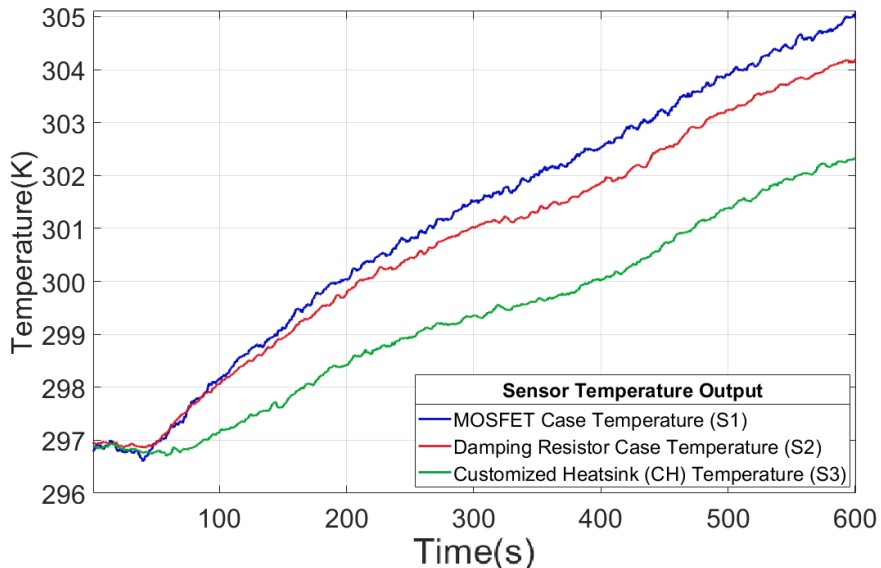

**Figure 6.** Temperature variation in ApsuTX with CH with 72 A coil current running for 10 minutes during the field test.

**Table 2.** Thermal performance and instrument parameters of ApsuTX with two different heatsinks.

| Performance Parameters for Extreme Test Conditions | Standard Heatsink (SH) | Customized Heatsink (CH) |
|---|---|---|
| Maximum Junction Temperature | 309 K (36°C) | 331 K (58°C) |
| Damping Resistance Temperature | 372 K (99°C) | 391 K (117°C) |
| Heatsink Surface Temperature | 308 K (35°C) | 330 K (57°C) |
| Heatsink Weight | 3.5 kg | 1.3 kg |
| Total ApsuTX Weight | 7.5 kg | 5.3 kg |
| Heatsink Dimensions (Length x Breadth x Thickness) | 30 cm x 20 cm x 4 cm | 30 cm x 20 cm x 0.8 cm |
| Total ApsuTX Dimensions | 30 cm x 20 cm x 19.8 cm | 30 cm x 20 cm x 16.6 cm |

obtained in the simulation with poor passive cooling, Fig. 5c. The sequences used in this test were one of the most extended high current sequences. Only a 5.3 K temperature rise in the CH is observed during the field test, which validates the CH

thermal performance well within safe operating conditions. Table 2 shows a comparison of all critical design parameters of the SH and CH. Both heatsinks perform satisfactorily for the extreme test conditions in the simulations. We note that the weight difference between SH and CH is 2.2 kg, which significantly enhances the compactness of ApsuTX without compromising the thermal performance. Also, 1920 cm$^3$ dimension reduction is achieved because of this modification. The total power dissipated by both heatsinks is 38 W, where the H-bridge dissipates 21 W, and the damping resistor dissipates 17 W. Figure 7 shows the

two ApsuTX systems.



**ApsuTx with Standard Heatsink(SH)**    **ApsuTx with Customized Heatsink(CH)**

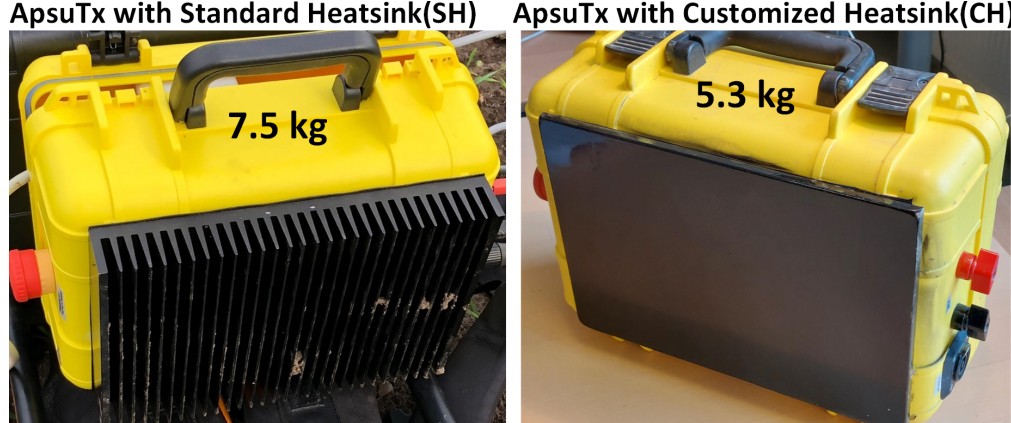

**Figure 7.** ApsuTX with standard heatsink (SH) and with the new customized heatsink (CH).

## 5.2 Current Drooping Results

In the drooping simulation, the time needed to stabilize the current output depends on the size of the capacitor bank. Figure 8a shows simulated inter-pulse drooping for different capacitor banks. The bank with 100 mF requires the longest time of about 75.5 s to stabilize due to its higher capacitance. Although the peak amplitude of the 100 mF bank is lower after stabilization, its

energy per pulse is high because of less intra-pulse drooping. The current pulse with 1 mF bank leads to maximum amplitude variation across each pulse because of higher intra-pulse drooping. We note that the amplitude of the pulses becomes more uniform with increasing capacitor bank size.

In steady-state sequences, the NMR signal is acquired between the pulses, and the signal directly depends upon the excitation pulse energy. We have evaluated the energy per pulse to understand the effect of intra-pulse drooping on the pulse energy.

Table 3 contains capacitance value with corresponding energy per pulse and pulse drooping time, i.e., duration required to stabilize the output. Ideally, the energy per pulse should be high, resulting in increased depth of investigation. The inter-pulse drooping must be removed for accurate groundwater exploration. Therefore, the pulse drooping time must be as small as possible to optimize the acquisition time and size of the capacitor bank. However, the parameters are inversely proportional, and both conditions can not be fulfilled simultaneously. Therefore, the selection of capacitance value is made based on optimum

energy per pulse and pulse drooping time. We have chosen 10 mF as a reasonable trade-off balancing the two parameters, and the form factor of the capacitor bank.

Figure 8b shows the actual drooping recorded during a survey. The observed drooping closely follows the simulation results. The 10 mF capacitor bank needs 3.6 s to stabilize in the simulation and it requires a similar time to stabilize in the Apsu instrument. For the example shown in Fig. 8b, the coil current is 70 A, and each pulse contains 27.99 J after drooping. As

discussed, achieving short drooping time and high pulse energy in the same compact and portable instrument is challenging. Still, with these component choices, we obtain a very useful trade-off. In conventional instruments, the capacitor bank can only

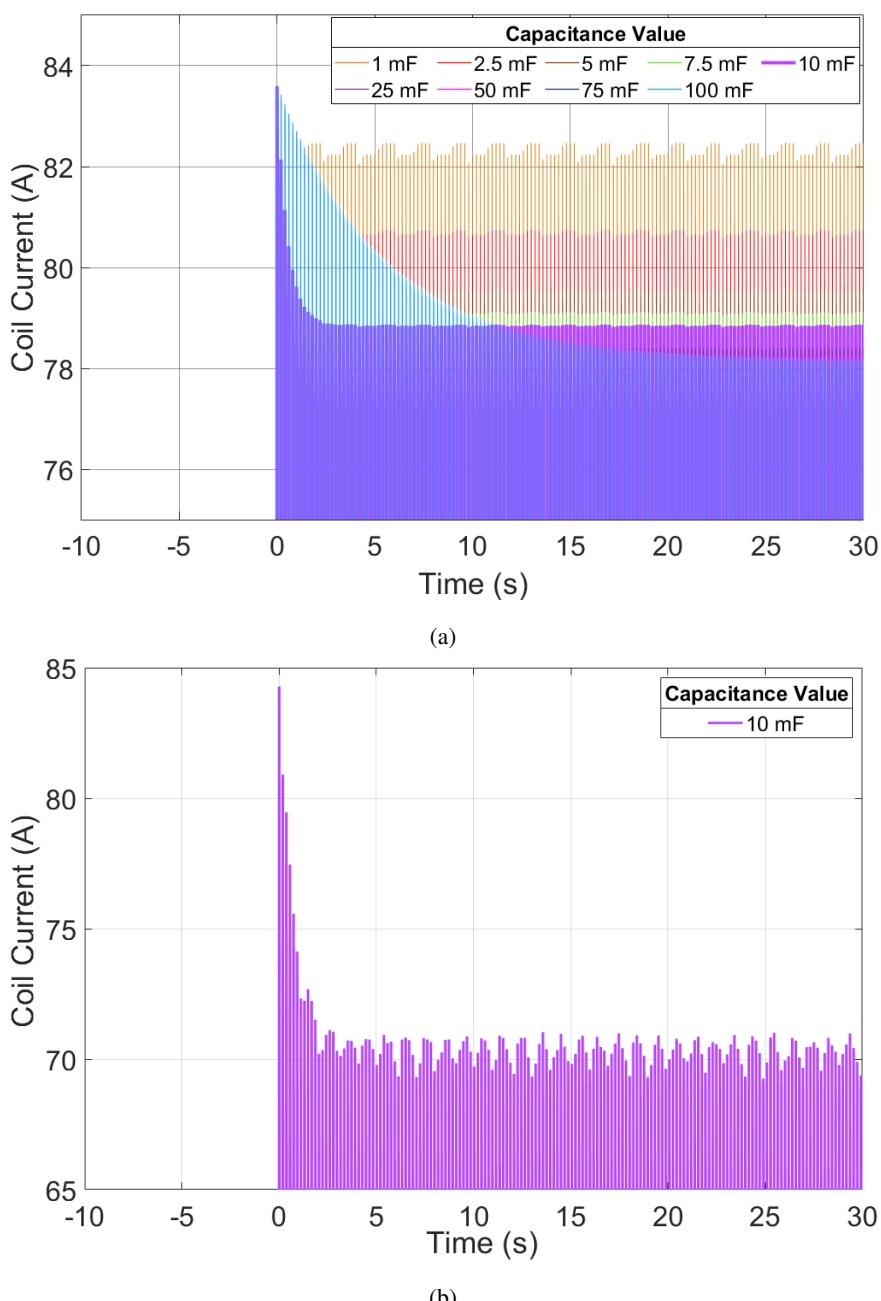

**Figure 8.** Drooping simulation and field results: a) Inter-pulse drooping in different capacitor banks in the simulation c) Inter-pulse drooping in the ApsuPS with 10 mF capacitor bank transmitting 70 A coil current in a field test.

be charged when the transmitter is off. However, ApsuPS and ApsuTx transmit NMR pulses and charge capacitor banks at the same time to maintain consistency in the pulse amplitude.



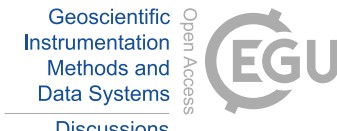

**Table 3.** Effect of bank capacitance value on the drooping time and energy per pulse after drooping in ApsuPS.

| Capacitance (mF) | Drooping Time(s) | Pulse Energy(J) |
|---|---|---|
| 1 | 1.8 | 34.9 |
| 2.5 | 1.8 | 37.7 |
| 5 | 1.8 | 38.2 |
| 7.5 | 2.8 | 38.3 |
| 10 | 3.6 | 38.4 |
| 25 | 7.4 | 38.4 |
| 50 | 16.6 | 38.4 |
| 75 | 26 | 38.4 |
| 100 | 37.2 | 38.5 |

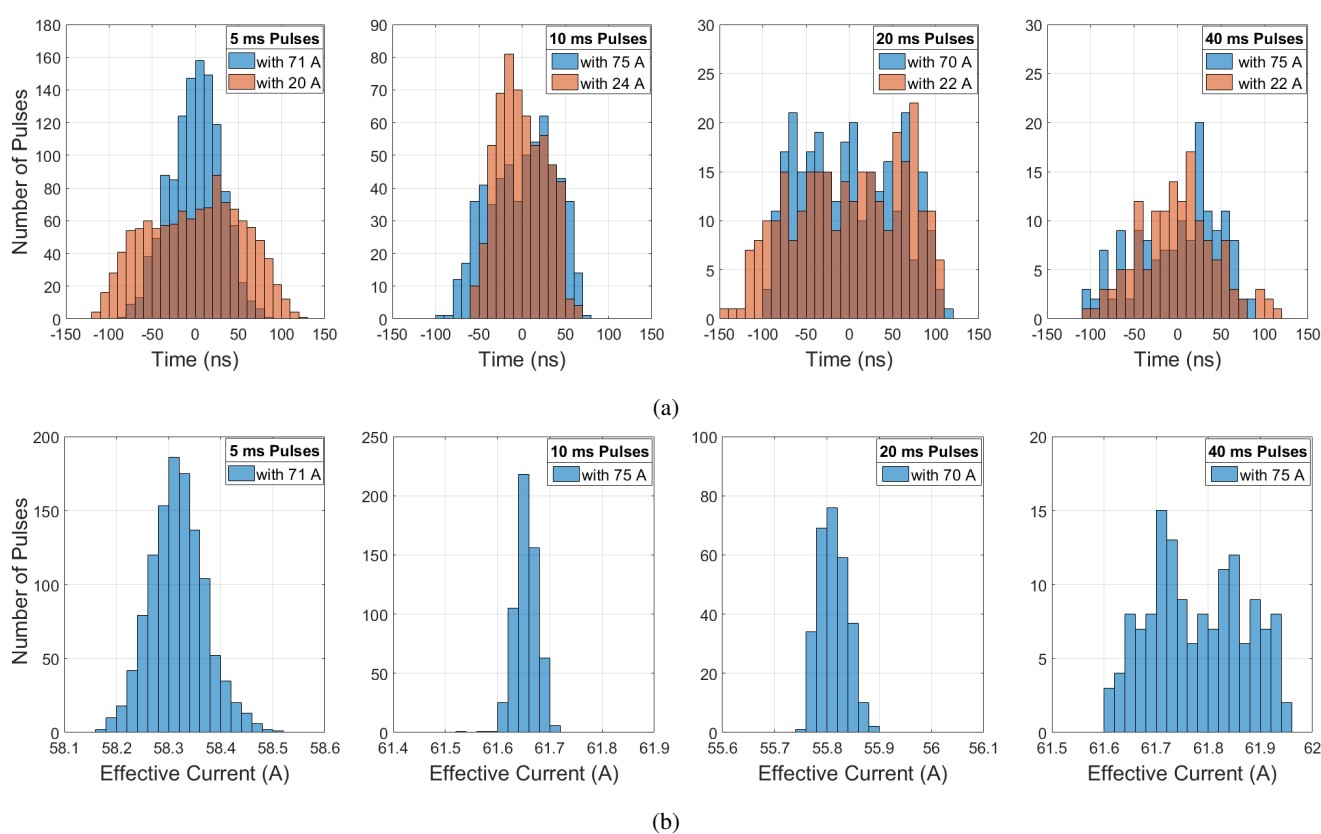

**Figure 9.** Pulse stability. Histograms of a) timing jitter and b) effective current jitter, between adjacent pulses in a 1 minute long sequence for different pulse lengths and currents. All sequences have a 10% duty cycle. The analysis has been done on data, where the initial drooping part of the pulse sequence has been removed.





### 5.3 Pulse Stability Results

The effectiveness of the steady-state NMR method depends crucially upon the pulse stability. The transmission frequency must be consistent across all sequences and the pulse phase and amplitude must be maintained uniformly throughout the sequences. We validate the stability of the ApsuTX transmitter by examining the generated pulse sequences. In Fig. 10 we show an eye diagram of the first oscillation in 1155 pulses from a pulse sequence with a amplitude of 75 A and 2.127 kHz Larmor frequency. The eye diagram is fully opened and demonstrates the high stability of the pulse train.

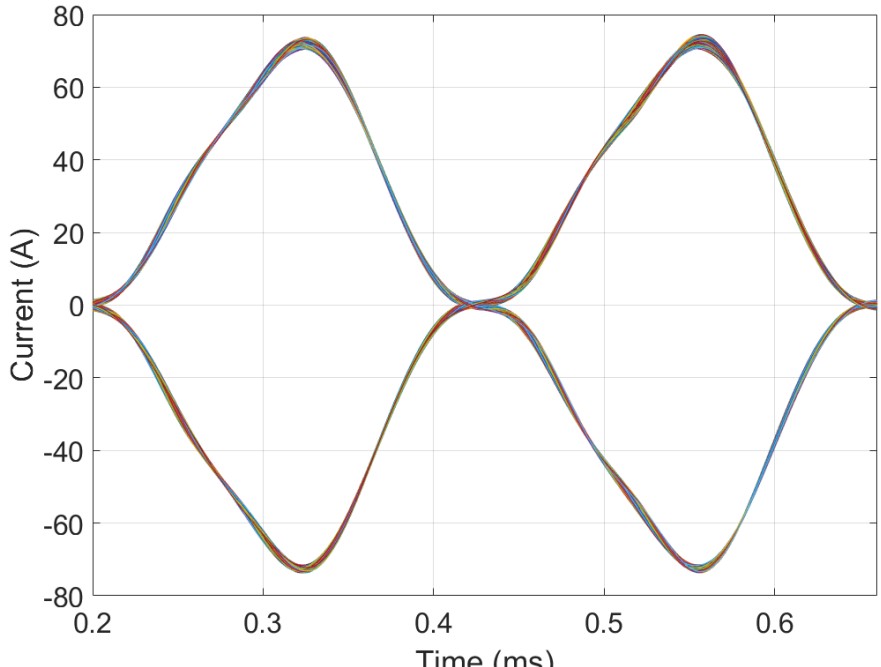

**Figure 10.** Eye diagram of 1155 transmitted pulses with 2.127 kHz Larmor frequency.

A different examination of the pulse train stability is given in Fig. 9. The data for this figure is generated by fitting a sinusoidal model at the Larmor frequency to each pulse in a 1 minute pulse sequence. From the models, we extract the pulse-to-pulse timing jitter and measure the pulse amplitude variations for two different currents and four different pulse lengths in Fig. 9a. The plots validated that the transmitter achieves less than $\pm 150$ ns timing jitter for all sequences and currents. This jitter corresponds to less than 0.22 degrees of phase error at Larmor frequency. Figure 9b shows the variations in the pulse currents

for the same sequences. The effective amplitude varies less than 0.5 A in all cases, which equals less than 0.4% variation. The average standard deviation in pulse amplitude and phase are 0.05 A and 42.6 ns respectively for the high currents. The plots verify that the ApsuTX transmits extremely stable pulses with consistent phase and amplitude throughout the pulse trains.

The increasing temperature of the transmitter during long measurements can potentially influence the pulse stability. To investigate this, we measured the H-bridge, damping resistor, and heatsink temperatures during continuous high-current trans-





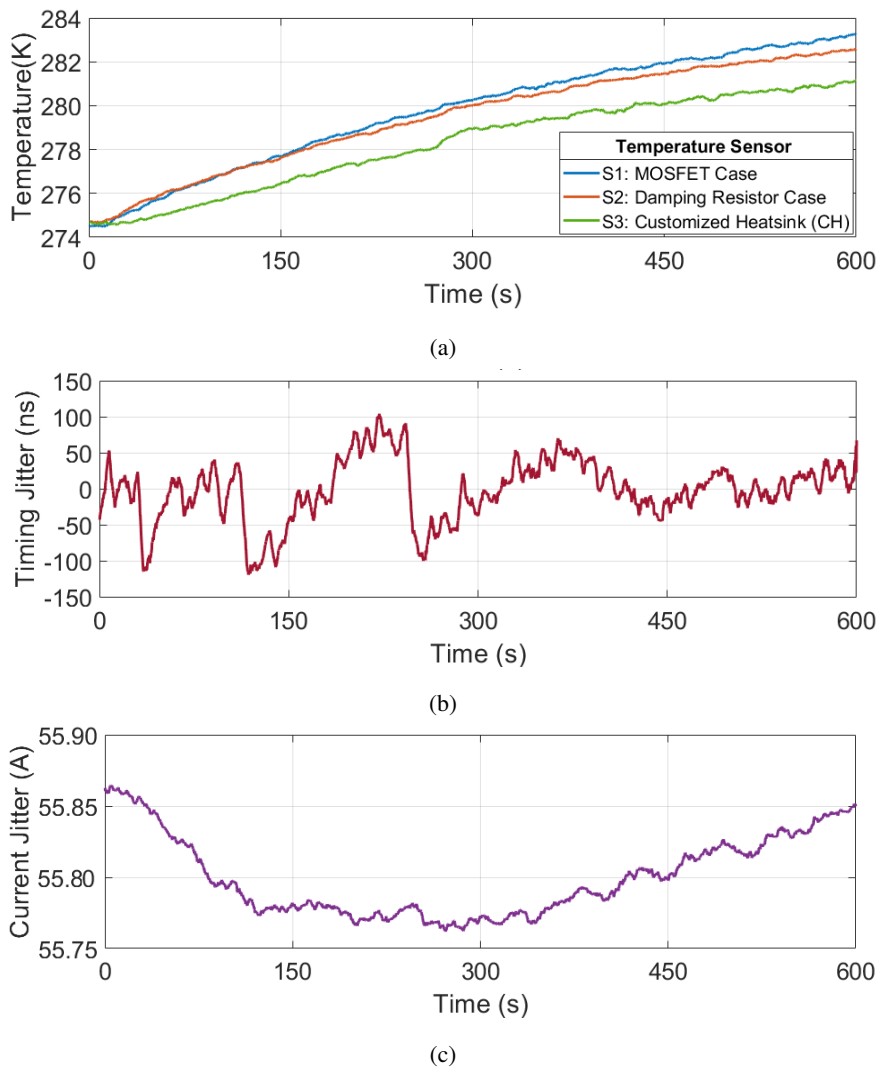

**Figure 11.** ApsuTx temperature changes, pulse timing jitter, and coil current jitter during a 10 minute sequence. a) change in MOSFET case temperature, damping resistor temperature, and customized heatsink temperature, b) timing difference between consecutive pulses, c) current amplitude for consecutive pulses. The measurement is based on 20 ms pulses and a 10% duty cycle.

mission of a 10 minute long pulse sequence of 20 ms pulses with a 10% duty cycle, Fig. 11. The MOSFET case and the damping resistor temperature case rise linearly by about 8 K, while the heat sink temperature rises by about 6 K during the measurement, Fig. 11a. Simultaneously, the timing jitter between pulses, Fig. 11b, and the pulse current amplitude, Fig. 11c, have been measured. The pulse-to-pulse jitter shows random variations between -150 ns and 150 ns without any noticeable systematic variation correlated with temperature changes in the ApsuTX sub-systems. Similarly, the current amplitude is consistently between 55.75 A and 55.90 A without any obvious correlation with temperature rise.

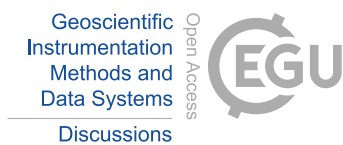

## 6 Discussion

The above results clearly demonstrate the ability of the new transmitter unit to generate highly stable pulse trains. Specifically, the observed pulse-to-pulse timing jitter and amplitude variations are so small, that the nuclear spin of hydrogen nuclei in groundwater can be efficiently driven into an NMR steady state. The steady-state transmitter unit has been efficiently used in numerous surveys in several countries. In this context it is noteworthy, that the 2.2 kg weight reduction obtained by the customized heatsink can seem like a small improvement, but it is highly appreciated by the field crews carrying the instrument between more than ten sites in a given day.

We typically use 60 s long pulse sequences in steady-state surface NMR measurements. The lower limit on the length of a pulse sequence is set by current drooping time and the time needed for the nuclear spins to reach the steady-state. This later time is controlled by the NMR relaxation times and is on the order of a few seconds for common geological materials. As the pulse sequence length is increased, the data are averaged over more repetitions and the signal-to-noise ratio is increased. Experimentally, we have found that a pulse sequence length of 60 s is a good comprise, where we obtain high signal-to-noise ratio data in a short amount of time. In practice many different pulse sequences with different current strength and pulse parameters are used and a full measurement employs about 30 sequences. At sites, where noise is a limiting factor, the signal-to-noise ratio can be improved using longer pulse sequences. The results presented in Fig. 6 show that after operating for 10 minutes at a high current of 72 A, the MOSFET case temperature has only increased by 8 K, well within safe limits. Consequently, it is possible to use very long pulse sequences if conditions necessitates so.

The depth of investigation in a surface NMR measurement is essentially controlled by the amplitude of the generated magnetic field. To obtain a deeper depth of investigation the amplitude of the magnetic field must be increased, either by using the same current in a larger transmitter coil or by using a larger current in the same size coil. However, the maximum current achievable is controlled by the ratio of the transmitter voltage and the coil self-inductance. Commercial surface NMR systems, employing FID measurements, are capable of depths of investigation down to about 100 m. To do so, transmitter voltages of several kV are used to generate 600-800 A currents in coils with a size of 100 m × 100 m. In contrast, the Apsu transmitter unit maximum current is ∼100 A in steady-state operation with a 50 m × 50 m coil and a 600 V supply, which translates to a depth of investigation of about 30 m. In principle, the extension of the steady-state concept to deeper depths of investigation through larger peak currents is simple, but the actual construction of a steady-state transmitter unit with outputs of several 100's of A is a significant engineering challenge left for future research.

## 7 Conclusion

In this article, the design and validation of a novel high duty cycle transmitter unit for steady-steady surface NMR measurements of groundwater was presented. The ability to generate a steady-state NMR response from groundwater hinges on the transmitter's ability to generate a high-fidelity train of coherent pulses with a negligible timing jitter and amplitude variation between pulses. In the article, we demonstrate that the transmitter unit has less than 150 ns pulse-to-pulse jitter and that the pulse amplitude is highly stable with less than 0.4% variation. Together, this makes the transmitter fully capable for steady-



state surface NMR measurements. The weight of the transmitter was optimized by designing a customized heatsink. Our results
demonstrate that compact and efficient steady-state surface NMR instruments are feasible. Our instrument is now in routine
use and have been collecting high SNR data at numerous sites. We foresee that future surface NMR instruments will all be
based on steady-state methods, in part or whole, due to demonstrated ability of this novel scheme to solve many of the issues
with poor signal-to-noise ratios.

*Acknowledgements.* This work was supported by Independent Research Fund Denmark under Grant 9041-00260B and VILLUM FONDEN
320 (35816).

*Competing interests.* The contact author has declared that none of the authors has any competing interests.

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
