# Peer review of "A High Duty Cycle Transmitter Unit for Steady-State Surface NMR Instruments"

_Geoscientific Instrumentation, Methods and Data Systems, 2023_

## Author Response (AR1)

**Dear Editor and reviewers**

Below is a list of the reviewer comment and how they have been addressed in the revised manuscript. Reviewer comments are in black, and our responses are in blue.

On behalf of the authors

Jakob Juul Larsen

**Reviewer 1 comments**

Main comment:

Under what environmental conditions were the field experiments conducted? Specifically, do you expect your customized heatsink to perform as effectively as the standard one in extreme conditions, such as during a heatwave?

The experimental data presented in the paper was acquired outside of the laboratory under calm conditions and an ambient temperature of about 15°C. The text has been modified and the sentence on measurement now reads (addition in italics):

"The CH has been fabricated and mounted on the updated ApsuTX and we have measured the temperature at three key test points during field tests *performed outdoors at about 15°C.*"

Since this manuscript was submitted, our instrument with the customized heatsink has been successfully used for extensive field work in, e.g., Senegal. Here, more than 100 soundings have been conducted at temperatures above 30 °C and more than 50 soundings have been conducted at temperatures above 40 °C. We can therefore now safely conclude that the performance of the customized heatsink is good. We have added a comment on our field experiences to the concluding remarks of the manuscript (addition in italics):

"Our instrument is now in routine use and has been collecting high SNR data at numerous sites including more than 100 soundings at temperatures above 30°C and more than 50 soundings at temperatures above 40°C."

**Specific comments**

1. notations on p.7:  $T_j$  and  $T_p$  are confusing. Use capital T for temperature  $(T_j)$  and t (t\_p) for time (even if it is a time period).

The manuscript has been revised according to these instructions.

2. The way to reference the figures is not homogeneous. Sometimes it is entitled "Fig." while other times it is "Figure". Authors need to check the manuscript. Also, the references are not actual active links.

The journal instructions states that: The abbreviation "Fig." should be used when it appears in running text and should be followed by a number unless it comes at the beginning of a sentence, e.g.: "The results are depicted in Fig. 5. Figure 9 reveals that..." We have checked the manuscript, and we have corrected the one instance where we depart from the requested formatting. Regarding the actual links in the manuscript, we have used the LaTeX bibliography style from the editor.

3. On Section 5, line 228, you state that "The temperature essentially follows the trend obtained in the simulation with poor passive cooling". What about the other cooling methods? Could you provide a comparison between the simulation and the actual measurements for both the standard Heatsink and the customized one? I believe that it would be interesting to add some simulations results on Figure 6 to support your arguments.

It is a good suggestion, but unfortunately, we did not record useable temperature data for the standard heatsink before it was replaced with the customized heatsink. We have expanded the text and added details on the similarities and dissimilarities between the simulations and the experiments.

The text has been rephased to: "The experimentally measured temperature rise essentially follows the trend obtained in the simulation with poor passive cooling of the CH. We find that the measured temperature increases by about 5 K after 600 s of pulsing at 72 A, which, given the expected uncertainty of the thermal model parameters, is in acceptable agreement with the simulated temperature rise of about 11 K after 600 s of pulsing at 93 A. The fact that only a 5 K temperature rise in the CH is observed during the extensive field test, validates the CH thermal performance and is well within safe operating conditions"

4. In Table 2, it is not clear if the values are measured ones or simulated ones. Could you explain how you obtained all critical design parameters? In particular, the values on this table are difficult to link with the results on Figure 5.

These are simulated values. We have updated the caption of Table 2:

"Simulated thermal performance of ApsuTX with the two different heatsinks at extreme test conditions pulsing at 93~A for 3000~s"

The headline "*Performance Parameters for Extreme Test Conditions*" has been changed to "*Parameter*"

Second, the wording used in the text, i.e., "critical design parameters" has been removed. The sentence now reads:

*"Table 2 shows a comparison of the simulated thermal performance of the two "heatsinks*

5. In Table 2, it would be important, for reproducibility purposes, to specify what "Extreme Test Conditions" are.

The extreme test conditions correspond to continuous pulsing at 93 A for 3000 s. In reality, we never pulse continuously at the highest current for more than 60 s. The text has been revised to:

"i.e., pulsing at the highest current for an extended period of time much longer than ever used in practice."

 Figure 8 is hard to read, and particularly Figure 8a. The negative time must be removed. As previously, adding the comparison between the measurement and simulation for 10mF (i.e. plotting the simulation with the measurements) would back up your statements.

The negative time has been removed. We have expanded the comparison between simulation with this change (addition in italics):

"Figure 8(b) shows the actual drooping recorded during a survey. The observed drooping closely follows the simulation results but are not identical due to limitations in the model, e.g., the model assumes an ideal DC supply feeding the capacitor bank."

7. Figure 10 is referenced before Figure 9.

This issue has been fixed.

8. Figure 9b is only showing the first intensity (~70A) compared to Figure 9a (~70A & ~20A).

This is a deliberate choice made to avoid having an excessive number of panels on the figure. The variations in the amplitude of the ~20 A pulses are similar to the variations in amplitude of the ~70 A pulses, so we only show the high amplitude results. We have added the following sentence:

*"For other currents (not shown), the standard deviations of pulse amplitude and timing are similar."*

**Reviewer 2 comments**

Specific comments:

Do you really need the TEM part in the introduction?

Yes and no. This paper is certainly not about TEM, but we have more and more field work where the acquisition of both TEM and NMR data allows us to draw much firmer conclusions than if data was collected with either TEM or NMR alone. We have therefore decided to keep the few lines on TEM to address this fact.

L.33: To me, relaxation time is only indirectly controlled by porosity. It is controlled by the pore size and mineral components of the pore walls. The amplitude is more closely connected to porosity because of: NMR amplitude -> (mobile) water content -> porosity

The text has been revised to "...relaxation time of the NMR signals is controlled by the host material, e.g., pore size and mineral composition,..."

L.57: Isn't the 5s argument not a bit stretched? This strongly depends on the T1 of your subsurface and I can rarely remember ever seeing T1 above 500ms in the field. So maybe it is more fair to link the experiment time to the T1 time of the subsurface?

The wait time is limited by the need for re-magnetization, but obviously this time is not known at the time of measurement. Further, the wait time can also be affected in some instruments by the need to recharge capacitors. The sentence has been revised to "…*a wait time of typically several seconds*…"

L.60-68: I totally agree on your point about the new steady-state approach and the massive SNR enhancement. Yes, this helps us to get better data to estimate water content (and therefore porosity) but still lacks the possibility to infer proper relaxation times, right? Or at least comparable to the standard FID measurements. So without the relaxation time, we only get half of the picture. Don't get me wrong, the half we are getting with steady-state is very good but a short hint/comment on the relaxation time issue would be nice (and objectively fair) I guess.

We would like to stress that steady-state surface NMR measures *both* amplitude and relaxation time. See for example the inversion results in this paper: M. P. Griffiths, D. Grombacher, L. Liu, M. Vang, and J. J. Larsen, Forward modelling steady-state free-precession in surface NMR, IEEE Transactions on Geoscience and Remote Sensing, DOI 10.1109/TGRS.2022.3221624, 2022.

We have added this reference along with the other references to steady-state surface NMR.

L.90: Maybe it is worthy to explain the term "drooping"? I have no electrical engineering background and I could basically infer from the context what it means. So maybe to an inexperienced reader a short explanation would be quite helpful?

This is a good point, and it has been addressed together with the next comment on Fig. 1.

Fig. 1: I don't know what the policy of the journal is but I personally like to have the figure explained to me in the text also and not just in the figure caption. I can live with it as it is but maybe this is a point to reconsider?

We have moved the majority of the caption text to section 2, "Challenges in steady-state surface NMR".

The moved text is slightly modified and now reads: "The defining aspect of the steady-state pulse train is that, first, phase coherence between pulses is maintained, i.e., timing jitter between pulses is neglible, and second, that all pulses are identical. The detrimental effect of timing jitter is illustrated in Fig.~1(b) where jitter prevents a steady-state from building up. Figure~1(c) demonstrates that steady-state operation is achieved with both identical boxcar shaped pulses as well as with identical pulses where the current amplitude drops by 25% within each pulse. In contrast, the results in Fig.~1(d) shows how current drooping between pulses again prevents a steady-state from building up."

L.125-L.130: You use the letter "T" for temperature  $(T_j)$  and time  $(T_p)$ . This is bad practice. Consider maybe using also the letter \tau for time related variables (with the obvious exception of T1 and T2) like you do in eq. 2.

This was also pointed out by reviewer 1 with an almost identical suggestion. We have updated the notation according to the reviewer 1 suggestion.

L.148: As you are already referring the Fig.8b (also a bit of bad practice), the caption says (c) instead of (b)

The reference to Fig.8b in line 148 is not essential and has been removed. The caption has been corrected.

Paragraph 5.1: I guess it is clear but you could explicitly mention that Fig. 5 shows only simulations.

This has been fixed. Please see the above response to reviewer 1 on this.

L.235: Technically, you could merge Fig.7 into Fig.4 so that the reader already there gets an impression of the decreased dimensions of the customized heatsink.

This is indeed a possibility, but we prefer to keep them as separate figure as they have different scope. Figure 4 is a picture of the experimental setup, whereas Figure 7 shows the "before and after" result. No changes have been made.

L.252: Description of Fig.8b: Can you comment on the variability of the pulse amplitude after the initial drooping. So from about 3s onward. Why does the intra-pulse drooping in the Apsu is much more pronounced compared to the simulations for 10mF (Fig.8a)?

We have added the following to address these questions (additions in italics).

"Figure 8(b) shows the actual drooping recorded during a survey. The observed drooping closely follows the simulation results but are not identical due to limitations in the model, e.g., the model assumes an ideal DC supply feeding the capacitor bank."

and

"Please note that the oscillations in the coil currents in Fig. 8 are not real, but sampling artefacts caused by the mismatch between the current sampling frequency and the pulse timing being controlled by the 2127 Hz Larmor frequency."

Paragraph 5.3: Maybe Fig.10 should come before Fig.9 as your explanation also starts with Fig.10?

This has been fixed.

L.316: I like the beginning of last sentence ("We foresee") but would rather take it with the wink of an eye.

It is indeed a speculative comment. Time will tell if we are right. No changes have been made to the paper.

**Technical corrections:**

- L.29 Earth's magnetic field (also in L.161)
- L.39 are of small amplitude or have a small amplitude
- L.43 have been very
- L.63 standard one
- L.159 contain instead of containing
- L.175 vary from
- L.316 and has been

These errors have been fixed.

---

## Referee Report (RR1)

**A High Duty Cycle Transmitter Unit for Steady-State Surface NMR Instruments**

The revised version of the manuscript, "A High Duty Cycle Transmitter Unit for Steady-State Surface NMR Instruments", has been carefully evaluated. The authors have adequately addressed the concerns raised in the previous review, notably regarding the figures and overall comprehension of the manuscript. The necessary revisions have been made to enhance the clarity and quality of the work.

All issues appear to have been resolved satisfactorily, and the manuscript is now considered suitable for publication. Acceptance is recommended.

---

## Author Response (AR2)

No comments for the accepted manuscript.